# Lower cancer incidence three years after COVID-19 infection in a large veteran population

**Jerry Bradley**[1], **Fei Tang**[2], **Natasha M. Resendes**[1,2], **Dominique M. Tosi**[1,2], **Iriana S. Hammel**[1,2]*

**1** Department of Medicine, University of Miami Miller School of Medicine, Miami, Florida, United States of America, **2** Miami Veterans Administration (VA)Healthcare System Geriatric Research, Education, and Clinical Center (GRECC), Miami, Florida, United States of America

* iriana.hammel@va.gov

## Abstract

### Background

The role of COVID-19 infection in cancer incidence risk is not known. COVID-19 infection may lead to increased cancer risk, as seen with other viruses, or to decreased risk due to the activation of the immune response during acute infection. This study aimed to determine the association between cancer incidence in US Veterans and COVID-19 infection.

### Methods

We conducted a retrospective cohort study of US Veterans comparing those who tested positive for COVID-19 during the first wave of COVID-19 between March 15, 2020, and Nov 30, 2020, to those who tested negative. We used data from the COVID-19 Shared Data Resource and Cox proportional hazard regression models to determine the hazard ratio of a new cancer diagnosis within a three-year follow-up period for the COVID-19 positive patients compared to those who were negative. Covariates included age, race, ethnicity, sex, BMI, smoking, being an active patient in the VHA system within a year of the COVID-19 test, and other factors.

### Results

499,396 patients were included in this study, with 88590 (17.2%) COVID-19 positive, 427566 (82.8%) COVID-19 negative. The ages of the COVID-19 positive and negative patients were $57.9 \pm 16.4$ and $59.5 \pm 15.8$, respectively. For those who survived for at least 30 days after COVID-19 testing, COVID-19 infection was associated with a 25% reduction in the hazard of cancer (HR = 0.75, 95% CI: 0.73–0.77). The reduction of the hazard was similar across sexes and races, except in Asians. Above 45 years of age, the hazard of cancer incidence further decreased with advancing age.

**Data availability statement:** We are not able to share the data directly as we obtained access to the U.S. VA national data through VA Informatics and Computing Infrastructure (VINCI). We signed an agreement that the data will not be downloaded and will stay at

the VINCI Workspaces behind VA firewall. Any request for the data can be sent to VINCI at VINCI@va.gov.

**Funding:** The authors declare no financial support was received for this study.

**Competing interests:** The authors have declared that no competing interests exist.

## Conclusions

Patients who were diagnosed with COVID-19 in the first wave of the pandemic had a decreased risk of cancer incidence in a 3-year follow-up across gender and race. Further multicenter prospective cohort studies are needed to evaluate the mechanism of this interaction.

## Introduction

The COVID-19 pandemic has introduced a new viral factor with relatively unknown effects on cancer development. Approximately 10% of cancers worldwide are virus-induced, with Epstein-Barr virus (EBV), Hepatitis C virus (HCV), and human papillomavirus (HPV) being examples of known viral oncogenic drivers [1]. The mechanism by which viruses lead to cancer development might be driven by the dysregulation of signaling pathways and alterations in host DNA and RNA processes [2–4]. It is unknown if, during the acute infection state of COVID-19, dysregulation in immune signaling [5,6], DNA damage with activation of pro-inflammatory processes [7], or other mechanisms might increase the risk of cancer development.

Early in the pandemic, researchers were aware that COVID-19 induced morphological and inflammatory changes in peripheral blood monocytes [8]. However, the significance of these changes has been debated. A recent analysis by Dr. Bharat et al. found that COVID-19 could induce non-classical monocytes to act with anticancer properties [9]. Specifically, they found that COVID-19 single-stranded RNA (ssRNA), unlike influenza ssRNA, could increase the population of this monocyte line. [9]. The implication of these changes as a function of monocyte exposure to COVID-19 ssRNA raises the question of whether contracting COVID-19 may affect cancer development in the short term.

Our team set out to compare the incidence of cancer in patients with COVID-19 to those without. We hypothesized that patients would have a reduced risk of cancer across the board in patients who were exposed to COVID-19 as compared to non-exposed patients.

## Methods

### Study design, data sources, and study population

We conducted a retrospective cohort study to assess the association between COVID-19 infection and new cancer diagnoses in Veterans who were free of cancer at baseline. We included Veterans who tested positive for the first time for SARS-CoV-2 (polymerase chain reaction or antigen test) in VA medical centers or clinics between March 15, 2020, and November 30, 2020. This time period encompasses the initial strain of COVID-19,before the emergence of the alpha and beta waves in December of 2020 [10,11]. The control group included Veterans who had a negative test result in this time frame and did not report positive test results in the next three years. We excluded patients who had any cancer diagnosis at baseline, those who died within 30 days after SARS-CoV-2 infection, and those who were not active

patients in the past 12 months (active defined as having at least one or more visits to a VA primary care clinic). We used nationwide data from the VHA medical centers from the VA COVID-19 Shared Data Resource. The Miami Veterans Affairs Healthcare System Institutional Review Board approved this study. The IRB granted a waiver for informed consent. The data was fully anonymized according to VHA standards prior to analysis. The data was accessed from October 1, 2024, to December 17, 2024. The study design and protocol were approved by the Miami VA Healthcare System Research and Development Committee (Reference: 1592780−10). Informed consent was waived as this is a data only research project.

### Study outcome

The primary outcome was any cancer diagnosis after COVID-19 testing (positive or negative) as documented by ICD-10 codes.

### Covariates

Age, race, ethnicity, sex, BMI, smoking, kidney disease, COPD, diabetes, hypertension, hyperlipidemia, obesity, chronic lung disease, PTSD, alcohol dependence, drug dependence, exposure to toxic substances, other viral infections such as a history of exposure to Hepatitis B, Hepatitis C, cytomegalovirus, mononucleosis, HIV among others, and immunodeficiency conditions secondary to prior history transplant or conditions related to immune response. COVID-19 severity was defined as following: mild as no hospitalization; moderate as hospitalized without ICU admission of high-flow oxygen; and severe as at least one of the following, high flow oxygen, intubation, mechanical ventilation, vasopressor, dialysis, ECMO.

### Statistical analysis

Continuous variables are presented as mean±standard deviation, median with interquartile range, and categorical variables as frequencies and percentages. When information on race, ethnicity and smoking status was not available, we reported the data as "Unknown." The Cox proportional hazard model was used to evaluate the association between COVID-19 and cancer diagnosis, adjusting for the covariates listed above. The proportional hazards assumption was assessed and verified by testing the correlation between the Schoenfeld residuals and survival time. Time-to-event analysis began 30 days after the date of infection, and patients were censored on the date of death or at the end of the years of follow-up. The association was also assessed in patients of different age groups, sexes, and races. Statistical analysis was performed using R (R project for statistical computing, version 4·0·5).

## Results

### Baseline characteristics

A total of 499,396 Veterans with SARS-CoV-2 testing in 2020 were included in this retrospective cohort study, with 334,709 (67.0%) white, 109,056 (21.8%) black, 42,870 (8.6%) Hispanic, 440,774 (88.3%) males. A total of 85,680 (17.2%) patients tested positive for SARS-CoV-2. The baseline characteristics of the study cohort are summarized in Table 1. The mean age of the entire cohort was 59.2±16.0 years (median 62 years), with COVID-19 positive and negative patients having similar ages (57.9±16.4 vs 59.5±15.8). The COVID-19 positive and negative patients were similar in terms of other baseline characteristics, as well.

### Association of COVID-19 infection and decreased cancer risk

Having a positive COVID-19 infection was associated with a 25% decrease in the hazard of having a cancer diagnosis in the three years follow-up (HR=0.75, 95% CI: 0.73–0.77), controlling for age, race, ethnicity, sex, BMI, smoking, kidney disease, COPD, diabetes, hypertension, hyperlipidemia, obesity, chronic lung disease, PTSD, alcohol dependence, drug

**Table 1. Baseline characteristics.**

| | Total<br>n = 499396 (100%) | COVID-19 Negative<br>n = 413716 (82.8%) | COVID-19 Positive<br>n = 85680 (17.2%) |
|---|---|---|---|
| Age, mean (years) ±SD (median; IQR) | 59.2 ± 16.0 (62; 48-71) | 59.5 ± 15.8 (62; 49-71) | 57.9 ± 16.4 (59; 46-71) |
| Age Groups, n (%) | | | |
| 19-64 | 285180 (57.1%) | 233200 (56.4%) | 51980 (60.7%) |
| 65 | 214216 (42.9%) | 180516 (43.6%) | 33700 (60.9%) |
| Male sex, n (%) | 440774 (88.3%) | 364774 (88.2%) | 76000 (88.7%) |
| Race, n (%) | | | |
| White | 334709 (67.0%) | 279421 (67.5%) | 55288 (64.5%) |
| Black | 109056 (21.8%) | 88774(21.5%) | 20282 (23.7%) |
| Asian | 6028 (1.2%) | 5207 (1.3%) | 821 (1.0%) |
| American Indian or Alaska Natives | 4246 (0.8%) | 3395 (0·8%) | 851(1.0%) |
| Native Hawaiian or Other Pacific Islander | 4633 (0.9%) | 3805 (0.9%) | 828 (1.0%) |
| Unknown | 40724 (8.2%) | 33114 (8.0%) | 7610 (8.9%) |
| Ethnicity, n (%) | | | |
| Hispanic | 42870 (8.6%) | 33197 (8.0%) | 9673 (11.3%) |
| Not Hispanic | 442912 (88.7%) | 369331 (89.3%) | 73581 (85.9%) |
| Unknown | 13614 (2.7%) | 11188 (2.7%) | 2426 (2.8%) |
| Smoking, n (%) | | | |
| Current | 106040 (21.2%) | 95505 (23.1%) | 10535 (12.3%) |
| Former Smoker | 186411 (37.3%) | 152985 (37.0%) | 33426 (39.0%) |
| Never | 163700 (32.8%) | 130262 (31.5%) | 33438 (39.0%) |
| Unknown | 43245 (8.7%) | 34964 (8.5%) | 8281 (9.7%) |
| Comorbidity, n (%) | | | |
| Kidney Disease | 98254 (19.7%) | 82089 (19.8%) | 16165 (18.9%) |
| COPD | 82247 (16.5%) | 71031 (17.2%) | 11216 (13.1%) |
| Diabetes | 148658 (29.8%) | 121548 (29.4%) | 27110 (31.6%) |
| Hypertension | 287591 (57.6%) | 239309 (57.8%) | 48282 (56.4%) |
| Chronic Lung Disease | 155285 (31.1%) | 131929 (31.9%) | 23356 (27.3%) |
| Overweight | 34270 (6.9%) | 26823 (6.5%) | 7447 (8.7%) |
| Toxic exposure | 3793 (0.8%) | 3219 (0.8%) | 574 (0.7%) |
| Alcohol Dependency | 119691 (24.0) | 101173 (24.5%) | 18518 (21.6%) |
| Drug Dependency | 34881 (7.0%) | 31236 (7.6%) | 3645 (4.3%) |
| Immunodeficiency | 14183 (2.8%) | 12122 (2.9%) | 2061 (2.4%) |
| Mortality | 84542 (16.9%) | 69520 (16.8%) | 14890 (17.4%) |

dependence, exposure to toxic substances, other viral infections, and immunodeficiency. The C-Index for the model was 0.733 (se = 0.001), indicating moderate predictive accuracy.

## Association of COVID-19 infection and decreased cancer risk by age groups, gender, and race

We assessed the association between positive COVID-19 infection and the risk of having a cancer diagnosis within three years in different age groups. We found that the older groups showed a greater reduction in the hazard of new cancer diagnosis. For the age group of 85−99, having a positive COVID-19 infection was associated with a 37% decrease in the hazard of having a cancer diagnosis in the three years follow-up (aHR: 0.63, 95% CI: 0.57−0.69), for age group of 75−84, the reduction was 27% (aHR: 0.73, 95% CI: 0.69–0.77); and for the age group of 55−64, the reduction was 17% (aHR:

0.83, 95% CI:0.79–0.88) (Table 2). Regarding sex, the reduction in new cancer diagnosis was slightly higher in men (aHR: 0.74, 95% CI: 0.73–0.76) than in women (aHR: 0.85, 95% CI: 0.77–0.95) (Table 3). Regarding race and ethnicity, the reduction in new cancer diagnoses was similar between white, black, Asian, and Hispanic populations (Table 4).

### Association of COVID-19 infection and decreased cancer risk by COVID-19 severity

When patients were stratified to three group according to COVID-19 severity, we showed that the reduction of newly diagnosed cancer risk was the most predominant in patients with mild COVID-19 (aHR = 0.72, 95% CI: 0.70–0.74) (Table 5). In patients with moderate COVID-19, the reduced cancer hazard was 11% (aHR = 0.89, 95% CI: 0.83–0.93). In patients with severe COVID-19, we did not show reduction in cancer hazard.

**Table 2.  aHR for new cancer diagnosis within three years for patients with positive COVID-19 testing compared to patients of negative testing by age. Adjusted for age, race, ethnicity, sex, BMI, smoking, kidney disease, COPD, diabetes, hypertension, hyperlipidemia, obesity, chronic lung disease, PTSD, alcohol dependence, drug dependence, exposure to toxic substances, other viral infections, and immunodeficiency.**

| Age groups | Number of events (%) | aHR (95% CI) |
| --- | --- | --- |
| 18-24 | 15 (0.5%) | 1.43 (0.44-4.58) |
| 24-34 | 369 (0.9%) | 0.86 (0.66-1.14) |
| 35-44 | 1090 (1.8%) | 0.93 (0.80-1.10) |
| 45-54 | 3221 (4.5%) | 0.85 (0.77-0.93) |
| 55-64 | 10728 (9.8%) | 0.83 (0.79-0.88) |
| 65-74 | 25185 (17.7%) | 0.76 (0.73-0.79) |
| 75-84 | 11204 (21.4%) | 0.73 (0.69-0.77) |
| 85-99 | 4028 (20.5%) | 0.63 (0.57-0.69) |

**Table 3.  aHR for new cancer diagnosis within three years for patients with positive COVID-19 testing compared to patients with negative testing by sex. Adjusted for age, race, ethnicity, sex, BMI, smoking, kidney disease, COPD, diabetes, hypertension, hyperlipidemia, obesity, chronic lung disease, PTSD, alcohol dependence, drug dependence, exposure to toxic substances, other viral infections, and immunodeficiency.**

| | Number of events (%) | aHR (95%CI) |
| --- | --- | --- |
| Man | 52729 (12.0%) | 0.74 (0.73-0.76) |
| Women | 3111 (5.3%) | 0.85 (0.77-0.95) |

**Table 4.  aHR for new cancer diagnosis within three years for patients with positive COVID-19 testing compared to patients of negative testing by race and ethnicity. Adjusted for age, race, ethnicity, sex, BMI, smoking, kidney disease, COPD, diabetes, hypertension, hyperlipidemia, obesity, chronic lung disease, PTSD, alcohol dependence, drug dependence, exposure to toxic substances, other viral infections, and immunodeficiency.**

| | Number of events (%) | aHR (95% CI) |
| --- | --- | --- |
| Black | 8183 (7.5%) | 0.78 (0.74-0.84) |
| White | 43326 (12.9%) | 0.74 (0.72-0.77) |
| Hispanic | 2570 (6.0%) | 0.76 (0.68-0.85) |
| Asian | 251 (4.2%) | 0.73 (0.47-1.15) |

**Table 5. aHR for new cancer diagnosis within three years for patients with positive COVID-19 testing compared to patients of negative testing by severity. Adjusted for age, race, ethnicity, sex, BMI, smoking, kidney disease, COPD, diabetes, hypertension, hyperlipidemia, obesity, chronic lung disease, PTSD, alcohol dependence, drug dependence, exposure to toxic substances, other viral infections, and immunodeficiency.**

|  | Number of events (%) | aHR (95% CI) | C-Index(se) |
|---|---|---|---|
| COVID-19 negative | 47614 (11.7%) | Reference | N/A |
| Mild | 5831 (7.8%) | 0.72 (0.70-0.74) | 0.735 (0.001) |
| Moderate | 1079 (12.6%) | 0.89 (0.83-0.93) | 0.729 (0.001) |
| Severe | 329 (11.9%) | 0.92 (0.82-1.02) | 0.729 (0.001) |

### Association of COVID-19 infection and decreased cancer risk by cancer type

The association of COVID-19 infection and different types of newly diagnosed cancer risk was accessed (Table 6). In all of the ten cancer types, we observed reduced hazard of having new cancer diagnosis in COVID-19 positive patients (Table 6).

## Discussion

### COVID-19 exposure and cancer risk

Our findings show that beyond the age of 45, exposure to COVID-19 is associated with a reduction in cancer incidence across all major cancer types. It is plausible that the immunologic changes that occur during acute COVID-19 infection [12,13] might lead to the activation of monocytes with anticancer properties, as has been reported by other investigators [9]. SARS-CoV-2 antigens induced increased expression of HLA-DR and TLR7 on monocytes compared to healthy cell controls [12]. In the context of cancer, HLA-DR and TLR7 play distinct but interrelated roles. HLA-DR, a class II MHC molecule, is involved in antigen presentation to CD4＋T cells, a crucial process for mounting an adaptive immune response against cancer. TLR7, a Toll-like receptor, is involved in recognizing single-stranded RNA, which can be found in viral infections and also in some cancer cells [13]. TLR7 can activate innate immune responses, potentially negatively influencing the tumor microenvironment and cancer cell behavior [14].The inflammatory response to COVID-19 may have a short-term protective benefit that wanes over time. Therefore, it is uncertain how long this risk reduction might be observed in

**Table 6. aHR for new cancer diagnosis within three years for male patients with positive COVID-19 testing compared to patients of negative testing by cancer types. Adjusted for age, race, ethnicity, sex, BMI, smoking, kidney disease, COPD, diabetes, hypertension, hyperlipidemia, obesity, chronic lung disease, PTSD, alcohol dependence, drug dependence, exposure to toxic substances, other viral infections, and immunodeficiency.**

|  | Number of events (%) | aHR (95% CI) |
|---|---|---|
| Prostate Cancer | 9037 (1.8%) | 0.81 (0.77-0.86) |
| Lung Cancer | 5594 (1.1%) | 0.63 (0.57-0.69) |
| Colorectal cancer | 2222 (0.4%) | 0.66 (0.58-0.76) |
| Kidney Cancer | 1851 (0.4%) | 0.80 (0.70-0.92) |
| Liver Cancer | 1801 (0.4%) | 0.69 (0.59-0.80) |
| Leukemias | 1261 (0.3%) | 0.84 (0.71-0.98) |
| Oral cancer | 1452 (0.3%) | 0.57 (0.48-0.68) |
| Pancreatic Cancer | 1271 (0.3%) | 0.71 (0.60-0.84) |
| Bladder Cancer | 2301 (0.5%) | 0.66 (0.58-0.75) |
| Melanoma | 2768 (0.6%) | 0.81 (0.72-0.90) |

the long term beyond the 3-year period of this study. Despite these possible mechanistic changes as a result of COVID-19 exposure, further research is required to explore the immunologic and physiologic properties of this potential association beyond those proposed in this paper. Care should be taken when interpreting these results, as these mechanisms are hypothetical and further definitive research is required.

## Pandemic healthcare disruptions

COVID-19 has introduced several new healthcare disruptions in cancer surveillance. Cancer trends during the pandemic showed an initial reduction, followed by a general return to the pre-pandemic levels [14,15]. One study looking at cancer screening rates among breast, cervical, and colorectal cancers found that screening initially dropped off during the start of the pandemic but resumed by the end of 2020, before experiencing a decline in 2021 [16]. An additional study showed that cancer incidence rates returned to their pre-pandemic trends in 2021, but those numbers did not fully incorporate delayed diagnosis from 2020 [17]. These changes were expected, given the drastic reduction in preventive services in the US [18]. An analysis of trends in US cancer rates revealed the highest variability in breast, cervical, prostate, and colorectal cancers, likely driven by a reduction in surveillance screening [19]. By 2023, breast and colorectal screening had improved from the pandemic-related declines, with cervical cancer screening remaining below pre-pandemic levels [20]. Additionally, the study collected data on all cancer types beyond those that are routinely screened for in clinical practice. It is uncertain how changes in cancer screening as described above, impacted the detection and treatment of these other cancer types. For this study, patients were followed for three years after the initial index infection. This included periods described above where the rates of screening for some cancer types returned to pre-pandemic levels or remained below expectations. Because both cohorts were exposed to similar screening delays and recovery over this period, it is unlikely that the difference in cancer rates between these groups could be attributed to this factor.

## Age and racial differences

Racial and sex disparities are known to exist after COVID-19 infection, with men being more likely to have adverse outcomes after infection [21]. These racial and gender disparities have improved over time [22] but remain an active area of investigation. Our findings showed that the overall rate of cancer reduction was the same regardless of race or sex, except for Asians, who comprised a small percentage of VA users. These findings were unexpected given the known racial and ethnic disparities in cancer incidence [23]. The relative uniformity of this reduction across both sexes and races suggests the potential of a biological process independent of these factors.

Regarding age, the incidence of cancer appeared to decrease with each decade of life in the COVID-19 group compared to that in the non-exposed group. This is surprising, given that cancer diagnoses typically increase with age. Several factors may contribute to this finding. Older patients have a higher rate of death than younger patients with COVID-19 [24]. Our study examined patients during the initial COVID-19 wave. During the pandemic, mortality rates reached their highest point in the delta-wave cohort [25]. We attempted to mitigate the risk of survival bias by censoring and multivariate analyses for potential confounders. Our study showed that mortality rates were similar between groups after the 30-day period. However, several factors remain that cannot be adjusted through statistical processes. Given the reduction in cancer rates across the board in the infected cohort, patients in this group who survived the initial infection may have been at a lower risk for cancer due to environmental or genetic reasons. Similarly, the increased mortality in this group could have conferred a survival advantage regarding cancer risk. Also, older Veterans may have been less likely to be screened for certain types of cancer and therefore less likely to be diagnosed during the follow up period. Care must be exercised when attempting to link viral exposure to reduced cancer outcomes, as the factors listed above may have influenced the outcomes Additionally, findings on race and age were within a Veteran-based cohort of predominantly older men. These findings may not be generalizable to other racial, age, gender, or ethnic groups.

### Impact of COVID-19 severity

Our study attempted to separate cancer risk based by COVID-19 severity of mild, moderate, and severe. Interestingly, patients with mild disease had the highest benefit, followed by those with moderate disease. No benefit was observed in patients with severe disease. Given that one of the proposed mechanisms is immunologic changes induced by the COVID-19 ssRNA [9], patients with severe disease may have experienced a higher dysregulated immunological response compared to milder disease presentations [26], possibly driven by higher levels of dysfunction between the adaptive and innate immune systems in severe cases of COVID-19 [27]. These changes in immunological response, which may prompt more severe disease progression, may reduce or eliminate any potential immunological changes that could aid in cancer surveillance. Further research is needed to investigate the degree of immune dysregulation in relation to COVID-19 severity and the development of inducible monocytes with cancer surveillance properties.

### Cancer types

Our study examined the top ten cancer types in men to determine if there was a primary driver behind the overall reduction observed. Surprisingly, no single cancer type was found to be influencing these results. All cancers showed various levels of reduction. These findings suggest a possible shared biological mechanism that might allow the immune system to identify cancer cells regardless of type. Further research is required to confirm the existence of this mechanism and how it influences the development of cancer.

### Strengths and limitations

Given the retrospective and observational nature of this study, the findings should be interpreted with caution. One of the strengths of this study is that it comprises a large nationwide database in which multiple risk factors and outcomes can be analyzed. An initial limitation of the study is that this is only a 3-year cohort analysis of cancer incidence. These findings might change as five and ten-year data become available and may be different in non-Veteran populations. Additionally, our study population (Table 1) was predominantly comprised of older men. These results may not be generalizable to other non-Veteran population, younger patients, or women.

We attempted to account for re-infection data; however, we were unable to capture all cases due to the increased availability of COVID-19 testing outside of the VA, such as at-home testing, state, and county testing sites, among others, and during subsequent re-infections. Veterans who may have had milder reinfection and did not seek treatment or testing, or other medical and socioeconomic factors, which may have limited testing, could not be accounted for beyond the initial study window. It is unknown how reinfection alters the risk of cancer development in these patients, and the extent to which it might induce other physiological changes. For our study, we separated the groups based on an initial index infection in which a veteran either tested positive or negative for COVID-19. Veterans were maintained in this initial grouping for the three-year follow-up period, even if those in the initial COVID-19 negative group may have subsequently developed an infection at a later period during the follow-up window or those in the initial group had reinfection. As a result, our findings may not accurately reflect the true effect size of this association. Regarding vaccination status, our initial study cohort was grouped prior to the availability of the COVID-19 vaccination and was maintained in this cohort status regardless of subsequent vaccination status during the follow-up period. It is uncertain how exposure from vaccination with mRNA might enhance or induce monocyte creation, similarly to that observed in the previously cited study [9]. In this study, inducible monocytes were only created when exposed to COVID-19 ssRNA and not that of influenza ssRNA. Further research is required to define potential mechanisms.

### Conclusion

Despite the limitations of our retrospective observational study, cancer incidence appears to be lower in patients diagnosed with COVID-19 across all major cancers and independent of race or sex. Further research is required to determine the precise causes and mechanisms underlying these findings.

## Supporting information

**S1 File. Human participants research checklist.**
(DOCX)

## Acknowledgments

We would like to acknowledge the support of the Miami VAMC, Geriatric Research Education and Clinical Center, and the US Veterans who served as our study population. We thank VINCI for their database that was provided to us, under VA Health Services Research Office No. 13–457. RES 13–457

## Author contributions

**Conceptualization:** Jerry Bradley.

**Data curation:** Fei Tang.

**Formal analysis:** Fei Tang.

**Investigation:** Jerry Bradley, Iriana Hammel.

**Methodology:** Jerry Bradley, Fei Tang, Iriana Hammel.

**Resources:** Fei Tang.

**Software:** Fei Tang.

**Supervision:** Iriana Hammel.

**Validation:** Jerry Bradley, Fei Tang, Natasha M. Resendes, Dominique M. Tosi, Iriana Hammel.

**Visualization:** Jerry Bradley.

**Writing – original draft:** Jerry Bradley.

**Writing – review & editing:** Jerry Bradley, Fei Tang, Natasha M. Resendes, Dominique M. Tosi, Iriana Hammel.

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
