## [Decision Letter · Decision Letter 0]

30 Apr 2025

PONE-D-24-59823
Lower Cancer Incidence Three Years After COVID-19 Infection in a Large Veteran Population
PLOS ONE

Dear Dr. Hammel,

Thank you for submitting your manuscript to PLOS ONE. After careful consideration, we feel that it has merit but does not fully meet PLOS ONE’s publication criteria as it currently stands. Therefore, we invite you to submit a revised version of the manuscript that addresses the points raised during the review process.

We look forward to receiving your revised manuscript.

Kind regards,

Anna D. Ware, MPH, MS

Academic Editor

PLOS ONE

Journal Requirements:

“This study was supported using resources and facilities of VINCI, VA HSR RES 13-457”

“This study was supported using resources and facilities of VINCI, VA HSR RES 13-457. The funder did not play any role in the study design, data collection and analysis, decision to publish or preparation of the manuscript.”

5. We note that you have indicated that there are restrictions to data sharing for this study. PLOS only allows data to be available upon request if there are legal or ethical restrictions on sharing data publicly. For more information on unacceptable data access restrictions, please see http://journals.plos.org/plosone/s/data-availability#loc-unacceptable-data-access-restrictions.  

Additional Editor Comments :

Dear authors,

Thank you for your valuable contributions to this important topic, which holds significant relevance for the scientific and clinical communities. Your study provides a meaningful evaluation of the association between COVID-19 infection and cancer incidence among Veterans, offering timely insights into a complex and evolving issue. Before we can proceed toward acceptance, the reviewers have suggested a major revision. Please find their feedback attached. In addition, I have outlined several suggestions below that I believe would help strengthen the clarity and robustness of your manuscript:

• Please provide a reference supporting the dates you assign to the Alpha and Beta variant waves (March 15–November 30, 2020).

• Consider separating Alpha and Beta wave periods if possible, as different variants are associated with differing clinical outcomes.

• Clarify how subsequent infections were handled. If a patient had multiple positive tests, which infection date was used to determine exclusion and outcomes?

• Expand on how “active patient in the past 12 months” was defined (e.g., minimum number of visits or types of care). Also, revise the description to avoid run-on structure.

• Describe how subsequent infections were treated in your time-to-event analyses beginning 30 days post-infection.

• Include model fit statistics (e.g., AIC, C-index, Schoenfeld residuals) to strengthen the robustness of your Cox model results.

• Reconsider whether “active patient status” should be an exclusion criterion rather than a covariate, given its implications for care utilization and potential external healthcare encounters.

Thank you for your work and patience while we obtained reviewers.

Sincerely,

Anna Ware, MPH, MS

Reviewers' comments:

Reviewer's Responses to Questions

**Comments to the Author**

1. Is the manuscript technically sound, and do the data support the conclusions?

Reviewer #1: Partly

Reviewer #2: Yes

2. Has the statistical analysis been performed appropriately and rigorously? 

Reviewer #1: Yes

Reviewer #2: Yes

3. Have the authors made all data underlying the findings in their manuscript fully available?

Reviewer #1: Yes

Reviewer #2: No

4. Is the manuscript presented in an intelligible fashion and written in standard English?

Reviewer #1: Yes

Reviewer #2: Yes

5. Review Comments to the Author

Reviewer #1: Of course there are three possibilities--increase, decrease or no effect. This is an interesting correlational finding, and even when you pull the confounds out a 25% difference remains, which is large enough to justify publication, but it could just be a statistical fluke.... particularly as all those people on the other side of the study had their immune systems stimulated by the COVID vaccine, as well as other corona and influenza viruses, so you are really positing this particular infection leads to a more robust protective response--including that of the vaccine.

Reviewer #2: This study presents an intriguing and timely investigation utilizing a large-scale cohort design, albeit with a relatively brief follow-up period. The topic carries substantial clinical relevance, and with appropriate revisions, the manuscript's impact could be considerably enhanced.

Methods

1)Study outcome:

The assertion that "screening returned to baseline" requires more robust empirical validation. Please incorporate specific quantitative data regarding outpatient encounters, screening utilization rates, or diagnostic procedural volumes in the post-pandemic period to substantiate this important claim.

Results

1)Line 101:

There is a numerical formatting error in "45,5388." Please rectify this typographical inconsistency.

2)Table 1:

Consider a more rigorous presentation of between-group differences. The inclusion of standardized mean differences (SMDs) would facilitate a more precise assessment of covariate balance following propensity matching or multivariate adjustment.

3)Association of COVID-19 Infection and Decreased Cancer Risk:

Lines 107-113 (Table 2)

Drawing substantive conclusions without stratification by cancer type is methodologically problematic. Given that various malignancies exhibit distinct pathophysiological mechanisms and screening paradigms, cancer-specific subgroup analyses would significantly enhance the interpretability of your findings and potentially elucidate specific protective associations.

4)Association of COVID-19 Infection and Decreased Cancer Risk by Age Groups, Gender, and Race:

Lines 120-130 (Table 4)

Given the predominance of male participants in your cohort, please acknowledge this as a potential limitation for generalizability and consider implementing gender-stratified analyses if statistically feasible.

5)Tables 2 and 4:

Please explicitly enumerate all confounding variables included in your statistical models as footnotes beneath the respective tables.

Discussion

1) COVID-19 Exposure and Cancer Risk

The proposed immune-mediated reduction in cancer risk appears speculative and insufficiently developed. Please expand this critical section with a more comprehensive mechanistic discussion grounded in current immunological literature, while clearly delineating the hypothetical nature of these postulated mechanisms.

2) Pandemic Healthcare Disruptions

Your follow-up period encompasses up to 3 years post-infection. Please precisely delineate what proportion of this observation period coincides with the normalization of cancer screening practices, and provide appropriate contextual evidence or references supporting this temporal framework.

3)Age and Racial Differences:

Given the distinctive attributes of the veteran population (regarding demographics, comorbidity profiles, and service history), please elaborate on how these characteristics might influence the external validity and generalizability of your findings to non-veteran populations.

4)Age and Racial Differences line 173-176:

The exclusion of individuals who succumbed within 30 days likely eliminates subjects with severe COVID-19 manifestations or compromised baseline health status. Would it be feasible to further stratify your cohort by COVID-19 severity indicators, such as hospitalization requirements or respiratory support needs, to explore differential cancer risk patterns?

6. PLOS authors have the option to publish the peer review history of their article (what does this mean?). If published, this will include your full peer review and any attached files.

Reviewer #1: **Yes: **Michael L. Russell

Reviewer #2: No

---

## [Author Response · Author response to Decision Letter 1]

5 Jun 2025

Please see our "Response to Reviewers" document.

---

## [Editor Report · Decision Letter 1]

11 Jul 2025

Lower Cancer Incidence Three Years After COVID-19 Infection in a Large Veteran Population

PONE-D-24-59823R1

Dear Dr. Hammel,

We’re pleased to inform you that your manuscript has been judged scientifically suitable for publication and will be formally accepted for publication once it meets all outstanding technical requirements.

Kind regards,

Anna D. Ware, MPH, MS

Academic Editor

PLOS ONE
---

## [Editor Report · Acceptance letter]

PONE-D-24-59823R1

PLOS ONE

Dear Dr. Hammel,

I'm pleased to inform you that your manuscript has been deemed suitable for publication in PLOS ONE. Congratulations! Your manuscript is now being handed over to our production team.

Kind regards,

on behalf of

Dr. Anna D. Ware

Academic Editor

PLOS ONE